# Completing State Representations using Spectral Learning

**Nan Jiang**
UIUC
Urbana, IL
nanjiang@illinois.edu

**Alex Kulesza**
Google Research
New York, NY
kulesza@google.com

**Satinder Singh**
University of Michigan
Ann Arbor, MI
baveja@umich.edu

## Abstract

A central problem in dynamical system modeling is state discovery—that is, finding a compact summary of the past that captures the information needed to predict the future. Predictive State Representations (PSRs) enable clever spectral methods for state discovery; however, while consistent in the limit of infinite data, these methods often suffer from poor performance in the low data regime. In this paper we develop a novel algorithm for incorporating domain knowledge, in the form of an imperfect state representation, as side information to speed spectral learning for PSRs. We prove theoretical results characterizing the relevance of a user-provided state representation, and design spectral algorithms that can take advantage of a relevant representation. Our algorithm utilizes principal angles to extract the relevant components of the representation, and is robust to mis-specification. Empirical evaluation on synthetic HMMs, an aircraft identification domain, and a gene splice dataset shows that, even with weak domain knowledge, the algorithm can significantly outperform standard PSR learning.

## 1 Introduction

When modeling discrete-time, finite-observation dynamical systems from data, a central challenge is state representation discovery, that is, finding a compact function of the history that forms a sufficient statistic for the future. Many models and algorithms have been developed for this problem, each of which represents and discovers state differently. For example, Hidden Markov Models (HMMs) represent state as the posterior over latent variables, and are learned via Expectation Maximization. Recurrent Neural Networks (RNNs) do not commit to any pre-determined semantics of state, but learn a state update function by back-propagation through time. Here, we focus on Predictive State Representations (PSRs), which represent state as predictions of observable future events [1, 2], and are unique in that they can be learned by fast, closed-form, and consistent spectral algorithms [3, 4].

Though they have been used successfully, spectral algorithms for PSRs attempt to discover the entire state representation from raw data, ignoring the possibility that the user has domain knowledge about what might constitute a good state representation. In many application scenarios, however, users do have such knowledge and can handcraft a meaningful, albeit incomplete state: for example, in many domains found in reinforcement learning [5, 6, 7], the last observation is often highly informative, and only a small amount of additional information needs to be extracted from the history to form a complete state. While spectral algorithms for PSRs are asymptotically consistent, ignoring domain knowledge and discovering state from scratch is wasteful and can result in poor sample efficiency.

In this work, we extend PSRs to take advantage of an imperfect, user-provided state function $f$, and design spectral algorithms for learning the resulting PSR-$f$ models. We theoretically characterize the relevance of $f$ to the system of interest, and show that a PSR-$f$ model can have substantially smaller size—and can thus be learned from less data—than the corresponding PSR. Our algorithm

computes principal angles to discover relevant components of $f$, and hence is robust to mis-specified representations. Experimental results show that this theoretical advantage translates to significantly improved performance in practice, particularly when only a limited amount of data is available.

## 2 Background

Consider a dynamical system $M$ that produces sequences of observations from a finite set $\mathcal{O}$ starting from some fixed initial condition. (The initial condition can be defined by a system restart or, if the system is not subject to restarts, it can be the stationary distribution.) For any sequence of observations $x \in \mathcal{O}^*$, let $P(x)$ be the probability that the first $|x|$ observations, starting from the initial condition, are given by $x$. Similarly, for any pair of sequences $h, t \in \mathcal{O}^*$, let $P(t|h) = P(ht)/P(h)$, where $ht$ denotes the concatenation of $h$ and $t$, be the probability that the next $|t|$ observations are given by $t$ conditioned on the fact that the first $|h|$ observations were given by $h$.

We say that $b : \mathcal{O}^* \to \mathcal{Z}$ is **state** for $M$ if $b$ is a sufficient statistic of history; that is, if all future observations are independent of past observations $h \in \mathcal{O}^*$ conditioned on $b(h)$. When the function $b(\cdot)$ is known, the system can be fully specified by $P(o|h) = P(o|b(h))$. And when $\mathcal{Z}$ is finite, the probabilities $P(o|z)$ for $o \in \mathcal{O}$ and $z \in \mathcal{Z}$ can be estimated straightforwardly from data.

In this paper we consider slightly more general state representations $b : \mathcal{O}^* \to \mathbb{R}^n$, letting $P(o|h) = \beta_o^\top b(h)$ for some $\beta_o \in \mathbb{R}^n$. This generalizes discrete-valued $b(\cdot)$ above because we can lift a discrete state to a one-hot indicator vector in $\mathbb{R}^n$ with $n = |\mathcal{Z}|$, in which case the $z$-th entry of $\beta_o$ is $P(o|z)$.

**PSRs** When $b(\cdot)$ is unknown, we need to learn both the state representation and $\{\beta_o\}_{o \in \mathcal{O}}$ from data. PSRs prescribe the state semantics:

$$b(h) = P_{\mathcal{T}|h} := [P(t|h)]_{t \in \mathcal{T}} \in \mathbb{R}^{|\mathcal{T}|} , \tag{1}$$

where $\mathcal{T} \subset \mathcal{O}^*$ is a set of appropriately chosen *tests*. Given a corresponding set of *histories* $\mathcal{H} \subset \mathcal{O}^*$, the matrix $P_{\mathcal{T},\mathcal{H}} := [P(ht)]_{t \in \mathcal{T}, h \in \mathcal{H}}$ plays a central role in PSR theory. $\mathcal{T}$ and $\mathcal{H}$ are called *core* sets if $P_{\mathcal{T},\mathcal{H}}$ has maximal rank, that is, $\mathrm{rank}(P_{\mathcal{T},\mathcal{H}}) = \mathrm{rank}(P_{\mathcal{O}^*,\mathcal{O}^*})$. We use $\mathrm{rank}(M)$ to denote that maximum rank, also known as the *linear dimension* of $M$ [2]. The linear dimension of an HMM, for example, is upper bounded by the number of latent states, regardless of the number of observations.

When $\mathcal{T}$ is core, $P_{\mathcal{T}|h}$ is provably a sufficient statistic of history, and there exist $\{\beta_o\}_{o \in \mathcal{O}}$ such that $P(o|h) = \beta_o^\top P_{\mathcal{T}|h}$ for all $h \in \mathcal{O}^*$. Furthermore, there exist updating matrices $\{B_o\}_{o \in \mathcal{O}}$ such that $P_{o\mathcal{T}|h} = B_o P_{\mathcal{T}|h}$, where $o\mathcal{T} = \{ot : t \in \mathcal{T}\}$. Knowing $\{B_o\}$ is sufficient to compute $P_{\mathcal{T}|h}$ for any $h \in \mathcal{O}^*$ by applying iterative updates to the initial state $b_* := P_{\mathcal{T}|\epsilon}$ ($\epsilon$ is the null sequence), since

$$P_{\mathcal{T}|ho} = \frac{P_{o\mathcal{T}|h}}{P(o|h)} = \frac{B_o P_{\mathcal{T}|h}}{\beta_o^\top P_{\mathcal{T}|h}}. \tag{2}$$

Altogether, we can compute $P(x)$ for any $x \in \mathcal{O}^*$ using a PSR $\mathcal{B} = \{b_*, \{B_o\}, \{\beta_o\}\}$.[1]

Thanks to the linear prediction rules, PSR parameters can be computed by solving linear regression problems $\{b(h) \mapsto P_{o\mathcal{T}|h} : h \in \mathcal{H}\}$ (for $B_o$) and $\{b(h) \mapsto P(o|h) : h \in \mathcal{H}\}$ (for $\beta_o$) [8], where each $h \in \mathcal{H}$ is a regression point and $\mathcal{H}$ being core guarantees that the design matrix $P_{\mathcal{T},\mathcal{H}}$ has sufficient rank. When $(\mathcal{T}, \mathcal{H})$ are core and $|\mathcal{T}| = |\mathcal{H}| = \mathrm{rank}(M)$ (in which case we say that $(\mathcal{T}, \mathcal{H})$ are *minimal core*), we have [9]:

$$b_* = P_{\mathcal{T}|\epsilon}, \quad B_o = P_{o\mathcal{T},\mathcal{H}}(P_{\mathcal{T},\mathcal{H}})^{-1}, \quad \beta_o^\top = P_{o,\mathcal{H}}(P_{\mathcal{T},\mathcal{H}})^{-1}, \ \forall o \in \mathcal{O}. \tag{3}$$

When only sample data are available, we can estimate the required statistics and plug them into Eq.(3), which yields a consistent algorithm.

## 3 PSR-$f$: Definitions and Properties

In this section we introduce the PSR-$f$, which extends the PSR by incorporating a "suggested" state representation $f : \mathcal{O}^* \to \mathbb{R}^m$ that is supplied by the user. Crucially, we show that when $f$ provides information relevant to the system, the PSR-$f$ model of the system can be more succinct than the PSR

model (Sec. 3.1). Succinctness often implies better finite sample performance, which is confirmed later by empirical evaluation in Sec. 6.

**State Representation** Given $f : \mathcal{O}^* \to \mathbb{R}^m$ and a set of tests $\mathcal{T}$, the state representation of a PSR-$f$, denoted by $b(h)$, is the concatenation of two components:

$$b(h) = \begin{bmatrix} P_{\mathcal{T}|h} \\ f(h) \end{bmatrix}. \tag{4}$$

While our formulation and results apply to arbitrary functions $f$, it is instructive to consider the special case $f(h) = P_{\mathcal{T}_f|h}$, where $\mathcal{T}_f$ is a set of $m$ user-specified independent tests, that is, $P_{\mathcal{T}_f,\mathcal{O}^*}$ has linearly independent rows.[2] In this case, we are essentially given a partial PSR state, and only need to find complementary tests $\mathcal{T}$ to complete the picture. In particular, $b(h)$ is a full state as long as $\mathcal{T} \cup \mathcal{T}_f$ is core, meaning that only $\mathrm{rank}(M) - m$ tests remain to be discovered. Similarly, if $f$ is lifted from a discrete-valued function (see Sec. 2), Appendix A shows that $f$ can often be viewed as a transformed predictive representation, making it natural to concatenate it with $P_{\mathcal{T}|h}$.

**Model Parameters and Prediction Rules** A PSR-$f$ has model parameters $\mathcal{B} = \{b_* \in \mathbb{R}^{m+|\mathcal{T}|}, \{B_o \in \mathbb{R}^{|\mathcal{T}| \times (m+|\mathcal{T}|)}\}, \{\beta_o \in \mathbb{R}^{m+|\mathcal{T}|}\}\}$. Below we specify the rules used to predict $P(o|h)$ from $b(h)$ and to update $b(h)$ to $b(ho)$. Using these rules, we can predict $P(x)$ for any $x \in \mathcal{O}^*$ in the same manner as standard PSRs (see Sec. 2).

**Prediction:** $P(o|h) \approx \beta_o^\top b(h)$. **State update:** $b(ho) = \begin{bmatrix} P_{\mathcal{T}|ho} \\ f(ho) \end{bmatrix}$, where $P_{\mathcal{T}|ho} \approx \dfrac{B_o b(h)}{\beta_o^\top b(h)}$.

(Note that we use approximate notation here because we have not yet characterized the conditions under which a PSR-$f$ will be exact.)

**Naïve Learning Algorithm** Recall that Eq.(3) can be seen as linear regression: $B_o$ is the solution to $\{b(h) \mapsto P_{o\mathcal{T}|h} : h \in \mathcal{H}\}$ and $\beta_o$ to $\{b(h) \mapsto P(o|h) : h \in \mathcal{H}\}$. We extend this idea to PSR-$f$.

Let $f_{\mathcal{H}}$ be a $m \times |\mathcal{H}|$ matrix whose $h$-th column is $f(h)$, and $P_{f,\mathcal{H}} := f_{\mathcal{H}} \mathrm{diag}(P_{\epsilon,\mathcal{H}})$; that is, its $h$-th column is $P(h)f(h)$. PSR-$f$ parameters can be computed by solving the following linear systems:

$$b_* = \begin{bmatrix} P_{\mathcal{T}|\epsilon} \\ f(\epsilon) \end{bmatrix}, \quad B_o \begin{bmatrix} P_{\mathcal{T},\mathcal{H}} \\ P_{f,\mathcal{H}} \end{bmatrix} \approx P_{o\mathcal{T},\mathcal{H}}, \quad \beta_o^\top \begin{bmatrix} P_{\mathcal{T},\mathcal{H}} \\ P_{f,\mathcal{H}} \end{bmatrix} \approx P_{o,\mathcal{H}}, \ \forall o \in \mathcal{O}. \tag{5}$$

For now we assume that $\begin{bmatrix} P_{\mathcal{T},\mathcal{H}} \\ P_{f,\mathcal{H}} \end{bmatrix}$ is invertible so that Eq.(5) can be solved by matrix inverse. This restriction will be removed in Sec. 4. When $m = 0$ we recover Eq.(3) for standard PSRs. Furthermore, by plugging in empirical estimates, we have a naïve algorithm that learns PSR-$f$ models from data.

The immediate next question is, when is this algorithm consistent?

## 3.1 Rank, Core, and Consistency

For PSRs, consistency requires core $\mathcal{H}$ and $\mathcal{T}$. Since PSR-$f$s generalize PSRs, we will need related but slightly different conditions for $\mathcal{H}$ and $\mathcal{T}$.

For the easy case where $f(h) = P_{\mathcal{T}_f|h}$, the answer is clear: Eq.(5) is consistent if $\mathcal{T} \cup \mathcal{T}_f$ and $\mathcal{H}$ are, respectively, minimal core tests and histories. This establishes that a PSR-$f$ model can be more succinct than a PSR model: if $\mathcal{T}_f$ consists of linearly independent tests, then with minmal $\mathcal{T}$ and $\mathcal{H}$ the size of each $B_o$ is $(\mathrm{rank}(M) - |\mathcal{T}_f|)\,\mathrm{rank}(M)$ for a PSR-$f$, compared to $\mathrm{rank}^2(M)$ for a PSR.

In the rest of this section, we extend the above result to the general case, where $f$ is an arbitrary function. We first give the definition of core tests/histories w.r.t. $f$, and establish consistency.

**Definition 1.** $(\mathcal{T}, \mathcal{H})$ *are core w.r.t.* $f$ *if*

$$\mathrm{rank} \begin{bmatrix} P_{\mathcal{T},\mathcal{H}} \\ P_{f,\mathcal{H}} \end{bmatrix} = \sup_{\mathcal{T}',\mathcal{H}' \subset \mathcal{O}^*} \mathrm{rank} \begin{bmatrix} P_{\mathcal{T}',\mathcal{H}'} \\ P_{f,\mathcal{H}'} \end{bmatrix}. \tag{6}$$

As with standard PSRs, we can consider core tests and histories separately; i.e., $\mathcal{T}$ (or $\mathcal{H}$) is core w.r.t. $f$ if there exists $\mathcal{H}$ (or $\mathcal{T}$) such that Definition 1 is satisfied.

**Theorem 1** (Consistency). *Solving Eq.(5) by matrix inverse is a consistent algorithm if $(\mathcal{T}, \mathcal{H})$ are core w.r.t. $f$ and $\begin{bmatrix} P_{\mathcal{T},\mathcal{H}} \\ P_{f,\mathcal{H}} \end{bmatrix}$ is invertible.*

The proof can be found in Appendix C. While Theorem 1 guarantees consistency, we have not yet illustrated the benefits of using $f$. In particular, we want to characterize the *sizes* of the minimal core tests/histories, as they determine the number of model parameters. At a minimum, we expect that $|\mathcal{T}| < \text{rank}(M)$ as long as $f$ is somewhat "useful". To formalize this idea, we introduce $\text{rank}(f; M)$ in Definition 5, and show that the minimal sizes of core $\mathcal{T}$ and $\mathcal{H}$ w.r.t. $f$ are directly determined by $\text{rank}(f; M)$ in Theorem 2. To get to those results, we first introduce the notion of linear relevance.

**Definition 2** (Linear Relevance). *$f$ is linearly relevant to $M$ if, for all $\mathcal{H} \subset \mathcal{O}^*$, $\text{rowspace}(P_{f,\mathcal{H}}) \subseteq \text{rowspace}(P_{\mathcal{T},\mathcal{H}})$[3], where $\mathcal{T}$ is any core set of tests for $M$.*

An interesting fact is that Definition 2 is equivalent to $f(h) = P_{\mathcal{T}_f|h}$ for some $\mathcal{T}_f$ up to linear transformations (see Prop. 2 in Appendix C). While $f$ may not be linearly relevant in general, we may expect that $f$ has some components that are linearly relevant, although it may also contain irrelevant information. To tease them apart, we introduce the following definitions.

**Definition 3.** *Define $\text{rank}(f) := \sup_{\mathcal{H} \subset \mathcal{O}^*} \text{rank}(P_{f,\mathcal{H}})$.[4]*

**Definition 4** (Linearly Relevant Components). *Let $U_f \in \mathbb{R}^{m \times n}$ be a matrix with the maximum number of columns $n$ such that (1) $f' := U_f^\top f(\cdot)$ is linearly relevant to $M$, and (2) $\text{rank}(f') = n$. For any $\mathcal{H} \subset O^*$, define $P_{f,\mathcal{H}}^\star := P_{f',\mathcal{H}}$.*

The matrix $U_f$ extracts the linearly relevant components from $f$, and our algorithm in Sec. 4 will learn such a matrix from data. Now we are ready to define $\text{rank}(f; M)$ and state Theorem 2.

**Definition 5.** *Define $\text{rank}(f; M) := \sup_{\mathcal{H} \subset \mathcal{O}^*} \text{rank}(P_{f,\mathcal{H}}^\star)$.*

**Theorem 2.** *The minimal sizes of $\mathcal{T}$ and $\mathcal{H}$ that are core w.r.t. $f$ are*

$$|\mathcal{T}| = \text{rank}(M) - \text{rank}(f; M), \qquad |\mathcal{H}| = \text{rank}(M) + \text{rank}(f) - \text{rank}(f; M).$$

The proof is deferred to Appendix C. The theorem states that, as expected, the higher $\text{rank}(f; M)$, the smaller $\mathcal{T}$. On the other hand, it also implies that the more irrelevant information $f$ contains, the larger $\mathcal{H}$ needs to be, which might seem counter-intuitive. Roughly speaking, this is because when $\mathcal{H}$ is small and $\text{rank}(f) - \text{rank}(f; M)$ is high, $f$ can have different behavior on $h \in \mathcal{H}$ and $h \notin \mathcal{H}$. The learning algorithm may be deceived by $f$'s good predictions on $\mathcal{H}$, only to find later that this does not generalize to new histories. In this case, we need to expand $\mathcal{H}$ to reveal $f$'s full behavior in order to have a consistent algorithm. A more concrete example on this issue can be found in Appendix C.1.

## 4   Spectral Learning of PSR-$f$s

One significant limitation of Eq.(5) for learning a PSR-$f$ is that the matrix $\begin{bmatrix} P_{\mathcal{T},\mathcal{H}} \\ P_{f,\mathcal{H}} \end{bmatrix}$ needs to be invertible, and finding $\mathcal{T}$ and $\mathcal{H}$ that satisfy that criterion can be difficult. In the PSR literature, this is known as the *discovery* problem, and is largely solved by *spectral learning* [4, 3]. Spectral algorithms, which are state-of-the-art for learning PSRs, take large $\mathcal{T}$ and $\mathcal{H}$ as inputs and then use singular value decomposition (SVD) to discover a *transformed* state representation $U_{\mathcal{T}}^\top P_{\mathcal{T}|h}$. In this section we devise spectral algorithms for learning PSR-$f$s; we will need to discover not only $U_{\mathcal{T}}$ as for traditional PSRs, but also the $U_f$ matrix that appears in Definition 4. The first step is to extend the PSR-$f$ formulation to allow transformed representations.

**Transformed PSR-$f$**   The state in a (transformed) PSR-$f$ is $b(h) = U_{\mathcal{T}}^\top P_{\mathcal{T}|h} + U_f^\top f(h)$, where $U_{\mathcal{T}} \in \mathbb{R}^{|\mathcal{T}| \times k}$, $U_f \in \mathbb{R}^{m \times k}$, and $k \leq |\mathcal{T}| + m$ is called the model rank. This representation

**Algorithm 1** Template for learning transformed PSR-$f$s

---

**Input:** $f : \mathcal{O}^* \to \mathbb{R}^m, U_{\mathcal{T}} \in \mathbb{R}^{|\mathcal{T}| \times k}, U_f \in \mathbb{R}^{m \times k}$.

1: $\hat{P}_{f,\mathcal{H}} := f_{\mathcal{H}} \operatorname{diag}(\hat{P}_{\epsilon,\mathcal{H}}). \quad U := \begin{bmatrix} U_{\mathcal{T}} \\ U_f \end{bmatrix}. \qquad \qquad \triangleright \hat{P}_{(\cdot)}$ is the empirical estimate of $P_{(\cdot)}$

2: $b_* := U_{\mathcal{T}}^\top \hat{P}_{\mathcal{T}|\epsilon}, \quad B_o := U_{\mathcal{T}}^\top \hat{P}_{o\mathcal{T},\mathcal{H}} \left( U^\top \begin{bmatrix} \hat{P}_{\mathcal{T},\mathcal{H}} \\ \hat{P}_{f,\mathcal{H}} \end{bmatrix} \right)^+, \quad \beta_o^\top := \hat{P}_{o,\mathcal{H}} \left( U^\top \begin{bmatrix} \hat{P}_{\mathcal{T},\mathcal{H}} \\ \hat{P}_{f,\mathcal{H}} \end{bmatrix} \right)^+.$

**Output:** $\mathcal{B} := \{b_*, \{B_o\}, \{\beta_o\}, U_f\}.$

---

**Algorithm 2** A basic spectral algorithm for PSR-$f$

---

**Input:** $f : \mathcal{O}^* \to \mathbb{R}^m$, model rank $k$.

1: $(U, \Sigma, V) := \operatorname{SVD}\left( \begin{bmatrix} \hat{P}_{\mathcal{T},\mathcal{H}} \\ \hat{P}_{f,\mathcal{H}} \end{bmatrix} \right). \qquad \qquad \triangleright$ singular values are in descending order

2: $U_{\mathcal{T}} := U_{1:|\mathcal{T}|, 1:k}. \ U_f := U_{(|\mathcal{T}|+1):(|\mathcal{T}|+m), 1:k}.$

**Output:** The output of Algorithm 1 on $f$, $U_{\mathcal{T}}$, and $U_f$.

---

generalizes Eq.(4), since we can recover the latter by letting $k = |\mathcal{T}| + m$ and

$$U_{\mathcal{T}} = \begin{bmatrix} \mathbf{I}_{|\mathcal{T}|}, \ \mathbf{0}_{m \times |\mathcal{T}|} \end{bmatrix}, \qquad U_f = \begin{bmatrix} \mathbf{0}_{|\mathcal{T}| \times m}, \ \mathbf{I}_m \end{bmatrix}, \tag{7}$$

where $\mathbf{0}$ and $\mathbf{I}$ are zero and identity matrices, respectively.

The parameters of a rank-$k$ transformed PSR-$f$ are $\mathcal{B} = \{b_* \in \mathbb{R}^k, \{B_o \in \mathbb{R}^{k \times k}\}, \{\beta_o \in \mathbb{R}^k\}, U_f \in \mathbb{R}^{m \times k}\}$. (Note that $U_{\mathcal{T}}$ is only used during learning, so it does not appear as a parameter.) After initializing $b(\epsilon) = b_* + U_f^\top f(\epsilon)$, the prediction and update rules are as follows.

$$\textbf{Prediction: } P(o|h) \approx \beta_o^\top b(h). \qquad \textbf{State update: } b(ho) \approx \frac{B_o \, b(h)}{\beta_o^\top \, b(h)} + U_f^\top f(ho).$$

**Template for spectral learning of a transformed PSR-$f$** If $k$, $U_f$, and $U_{\mathcal{T}}$ are given, we can easily adapt the algorithm in Eq.(5) to compute the model parameters of a transformed PSR-$f$. See Algorithm 1, where $(\cdot)^+$ is the matrix pseudo-inverse. In the rest of this section, we will introduce spectral algorithms that use Algorithm 1 as a subroutine and differ in their choices of $U_{\mathcal{T}}$ and $U_f$.

### 4.1 A simple algorithm

The key operation in spectral algorithms for standard PSRs is the SVD of $P_{\mathcal{T},\mathcal{H}}$ [3, 4]. The analog of $P_{\mathcal{T},\mathcal{H}}$ in our setting is $\begin{bmatrix} P_{\mathcal{T},\mathcal{H}} \\ P_{f,\mathcal{H}} \end{bmatrix}$, so our first algorithm simply takes the SVD of $\begin{bmatrix} P_{\mathcal{T},\mathcal{H}} \\ P_{f,\mathcal{H}} \end{bmatrix}$ to obtain $U_{\mathcal{T}}$ and $U_f$; see Algorithm 2. Note that the standard spectral algorithm is recovered when $m = 0$. Algorithm 2 is consistent under certain conditions; the proof is deferred to Appendix E.

**Theorem 3.** *Given any $f$, Algorithm 2 is consistent when $\mathcal{T}$ and $\mathcal{H}$ are core w.r.t. $f$ and $k = \operatorname{rank}(M) + \operatorname{rank}(f) - \operatorname{rank}(f; M)$.*

Despite its consistency, the algorithm has some significant caveats. In particular, Theorem 3 implies that we may need $k > \operatorname{rank}(M)$ to guarantee consistency, which *increases* the state dimensionality. To see why this is inevitable, consider the scenario where $\|f(h)\| \gg \|P_{\mathcal{T}|h}\|$. $P_{f,\mathcal{H}}$ will dominate the spectrum of $\begin{bmatrix} P_{\mathcal{T},\mathcal{H}} \\ P_{f,\mathcal{H}} \end{bmatrix}$, and the first $\operatorname{rank}(f)$ singular vectors will likely depend on $P_{f,\mathcal{H}}$, *regardless of whether $f$ is relevant or not*. Since the algorithm picks singular vectors based only on their singular values, we are forced to keep irrelevant components of $f$ in our state representation, causing a blow-up in dimensionality.

### 4.2 Identification of linearly relevant components by principal angles

Ideally, we would like to identify the linearly relevant components of $f$ (Definition 4) and discard the irrelevant parts. If we had access to exact statistics $P_{\mathcal{T},\mathcal{H}}$, we could identify those linearly relevant

---

**Algorithm 3** Canonical angle algorithm for PSR-$f$

---

**Input:** $f : \mathcal{O}^* \to \mathbb{R}^m$, model rank $k$, $0 \le d \le \min(k, m)$.

1: $(U'', \Sigma'', V'') := \text{SVD}(\hat{P}_{\mathcal{T}, \mathcal{H}})$.
2: $\tilde{P}_{f, \mathcal{H}} := \hat{P}_{f, \mathcal{H}}$, row-orthonormalized, $\tilde{P}_{\mathcal{T}, \mathcal{H}} := U''^{\top}_{1:|\mathcal{T}|, 1:k} \hat{P}_{\mathcal{T}, \mathcal{H}}$, row-orthonormalized.
3: $(U_a, \Sigma_a, V_a) := \text{SVD}(\tilde{P}_{\mathcal{T}, \mathcal{H}} \tilde{P}_{f, \mathcal{H}}^{\top})$.      ▷ compute principal angles
4: $(U')^{\top} := ([V_a]_{(:), 1:d})^{\top} \tilde{P}_{f, \mathcal{H}} (\hat{P}_{f, \mathcal{H}})^{+}$.
5: $f'(\cdot) := \lambda \cdot (U')^{\top} f(\cdot)$ for some large $\lambda \in \mathbb{R}$.
6: $\{b_*, \{B_o\}, \{\beta_o\}, U_{f'}\} :=$ the output of Algorithm 2 on $f'$ and $k$.

**Output:** $\mathcal{B} := \{b_*, \{B_o\}, \{\beta_o\}, U'U_{f'}\}$.      ▷ $U' \in \mathbb{R}^{m \times d}$, $U_{f'} \in \mathbb{R}^{d \times k}$

---

components by computing the *principal angles* between the row space of $P_{\mathcal{T}, \mathcal{H}}$ and that of $P_{f, \mathcal{H}}$ [10]. Define $\tilde{P}_{\mathcal{T}, \mathcal{H}}$ to be a matrix whose rows form an orthonormal basis of $P_{\mathcal{T}, \mathcal{H}}$, and $\tilde{P}_{f, \mathcal{H}}$ similarly for $P_{f, \mathcal{H}}$. The singular values of $\tilde{P}_{\mathcal{T}, \mathcal{H}} \tilde{P}_{f, \mathcal{H}}^{\top}$ correspond to the cosine of their principal angles. In particular, if the intersection of the row spaces of $P_{\mathcal{T}, \mathcal{H}}$ and $P_{f, \mathcal{H}}$ is $d$-dimensional, then the first $d$ singular values will be $\cos(0) = 1$, and the remaining singular values will be less than $1$.

When we only have access to empirical statistics and/or $f$ only approximately contains linearly relevant components, the leading singular values will be close to but less than $1$. Based on this observation, Algorithm 3 computes principal angles and extracts the relevant components of $f$ in a way that is robust to statistical noise. Line 1 uses the standard spectral learning procedure to compress $\hat{P}_{\mathcal{T}, \mathcal{H}}$ and remove the dimensions that correspond to pure noise. Lines 2 and 3 compute the principal angles via SVD. Line 4 uses the right singular vectors to extract the $d$ most relevant dimensions from $f$. And, finally, the last line calls Algorithm 2 with a new function $\lambda \cdot (U')^{\top} f$ that only contains the identified relevant components.

**Preservation of dimensionality and invariance to transformations**  A consistency guarantee for Algorithm 3 is stated below; it shows that the dimensionality of learned state will not blow up. Furthermore, by design the algorithm is *invariant to transformations*, in the sense that $f$ and $A^{\top} f$ will produce the same result for any full-rank matrix $A$, thanks to the orthonormalization step (Line 2).

**Theorem 4.** *Given any $f$, Algorithm 3 is consistent as long as $(\mathcal{H}, \mathcal{T})$ is core w.r.t. $f$, $k = \text{rank}(M)$, $d = \text{rank}(f; M)$, and $\lambda$ is a fixed positive constant.*

$\lambda \to \infty \Rightarrow$ **Reduced model complexity**  In Sec. 3.1 we saw that when $f$ is linearly relevant, $B_o$ for a non-transformed PSR-$f$ only needs $\text{rank}(M)(\text{rank}(M) - \text{rank}(f; M))$ parameters. In a transformed PSR-$f$, however, the size of $B_o$ is *always* $k \times k$. To guarantee consistency we need $k \ge \text{rank}(M)$, so at a superficial level no savings in model parameters seems possible.

However, when we re-express a non-transformed PSR-$f$ in the transformed form (Eq.(7)), the $U_{\mathcal{T}}$ matrix has many zero entries, which leads to zeros in $B_o \in \mathbb{R}^{k \times k}$ (see Algorithm 1), implying that the effective size of $B_o$ can be much smaller than $k^2$. Based on this observation, we show that when $\lambda \to \infty$, Algorithm 3 produces $B_o$ that has at most $k(k - d)$ non-zero entries.

**Proposition 1.** *In the limit as $\lambda \to \infty$, $B_o$ has at most $k(k-d)$ non-zero entries for all $o$. When $k$ and $d$ are as in Theorem 4, the number of non-zero entries is at most $\text{rank}(M)(\text{rank}(M) - \text{rank}(f; M))$.*

See the proof and additional details in Appendix E.1. When $k$ and $d$ are as in Theorem 4, the effective size of $B_o$ matches the analysis in Sec. 3.1. Notably, unlike in Sec. 3.1, Prop. 1 does *not* rely on $f$ being linearly relevant, and the model complexity is as if only the linearly relevant components of $f$ were given to begin with.[5] In practice, the algorithm behaves robustly for any reasonably large $\lambda$.

## 5  Related Work

Our work has a similar motivation to that of James et al. [5] (and related work [6, 11]), who incorporate a user-provided partition on PSR histories by learning one model for each partition; these are called

mPSRs. In our setting, a partition over histories can be represented as an $f$ that includes indicators of partition membership[6] and our results apply more generally to any real-valued function. While they show examples where mPSRs can have fewer parameters than PSRs, we are not aware of a general characterization of when this happens; the strongest existing result is that the model size will not blow up (see Theorem 1 in James et al. [5]). In contrast, we are able to characterize the relevance of an arbitrary function $f$ and quantify the model size explicitly (Sec. 3.1 and Prop. 1).

Feature PSRs [4] also attempt to leverage domain knowledge by using user-provided history and test features to improve learning efficiency. While our use of $f$ is superficially similar, in fact the two approaches are quite distinct (and complementary). In our formulation, $f$ forms a part of the state that is computed directly and not maintained via iterative updates. In contrast, for feature PSRs, all dimensions of the state still need to be discovered from data and updated iteratively during prediction. Further discussion of these differences can be found in Appendix D.

## 6 Experiments

### 6.1 Synthetic HMMs

**Domain** We generate HMMs with 10 states and 20 observations as the ground truth. Each state has 3 possible observations and 3 possible next states. We consider two types of topologies: with RAND topology, the possible next states for each state are chosen uniformly at random; with RING topology, the states form a ring and each state can only transition to its neighbors or itself. All non-zero parameters of the HMMs are generated by sampling from $U[0, 1]$ followed by normalization.

**The function** $f$ For each HMM we provide two functions: the first function ("dummy-0") takes the form of $f(h) = P_{\mathcal{T}_f | h}$, with $\mathcal{T}_f$ containing 3 independent tests. The second function ("dummy-3") appends 3 more features to the first one. The new features are predictions of 3 independent tests but for a different HMM[7] hence are irrelevant to the HMM of interest. While we might want to make the problem more challenging by transforming the function so that the relevant and the irrelevant features mix together, this has no effect on Algorithm 3 since it is invariant to transformation (Sec. 4).

**Algorithm details** For both standard spectral learning ("PSR") and Algorithm 3 ("PSR-f-dummy-X"), $\mathcal{T}$ and $\mathcal{H}$ consist of all the observation sequences of lengths 1 and 2. The hyperparameter $d$ for Algorithm 3 is tuned by 3-fold cross validation on training data, and $\lambda$ is set to 100 to ensure a succinct model (see Appendix E.1). We additionally include a baseline that uses $f$ (without useless features) as state and only learns vectors $\{\beta_o\}$ such that $P(o|h) \approx \beta_o^\top f(h)$ ("f-only").

**Results** From each HMM we generate 500, 1000, and 2000 sequences of length 5 as training data. The models are evaluated by the log-loss (i.e., negative log-likelihood) on 1000 test sequences of length 5. Fig. 1a shows the log-loss of different algorithms as a function of model rank $k$ for sample size 1000, and the rest of the results can be found in Appendix F. We can see that PSR-$f$ models outperform PSRs across all model ranks and all sample sizes. While using $f$ alone gives somewhat comparable performance in the low sample regime, it fails to improve with more data due to its incomplete state representation, whereas PSR-$f$ can leverage an imperfect $f$ while remaining consistent. Finally, while adding irrelevant features hurts performance in the small sample regime, the degradation is only mild and goes away as more data become available.

### 6.2 Aircraft Identification

The next domain is a POMDP developed by Cassandra [12], which we convert into an HMM by using a uniformly random policy. The POMDP simulates a military base using noisy sensors to decide whether an approaching aircraft is friend/foe, and how far away it is. See Appendix F and [12, Chapter H.4] for a detailed specification. Each observation consists of a binary foe/friend signal and a distance, both of which are noisy. We average both components over all previous time steps to obtain $\hat{e}$ and $\hat{l}$, respectively—intuitively these should be relevant to the state—and compute

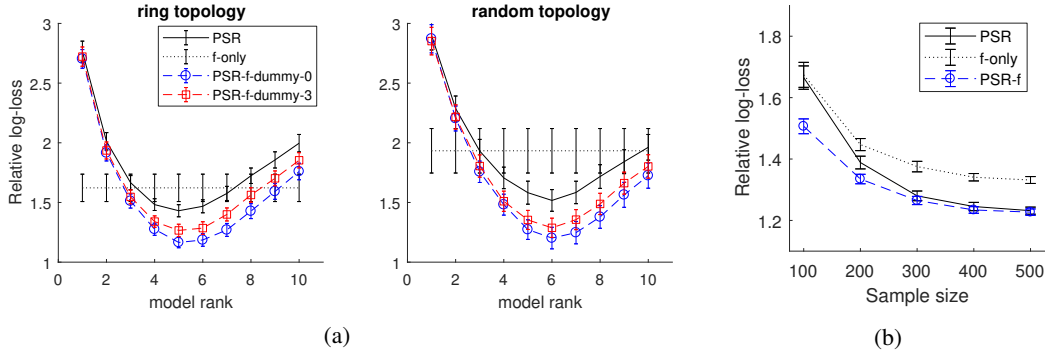

(a)                                                                                              (b)

Figure 1: **(a)** Synthetic HMMs (Sec. 6.1). The $y$-axis is relative log-loss (the lower the better), where zero corresponds to the log-loss of the ground truth model. "f-only" does not depend on model rank and is a horizontal line. All results are averaged over 100 trials, and all error bars in this paper show 95% confidence intervals. Sample size is 1000. See text and Appendix F for more details and full results. **(b)** Aircraft Identification domain (Sec. 6.2). Results are averaged over 100 trials.

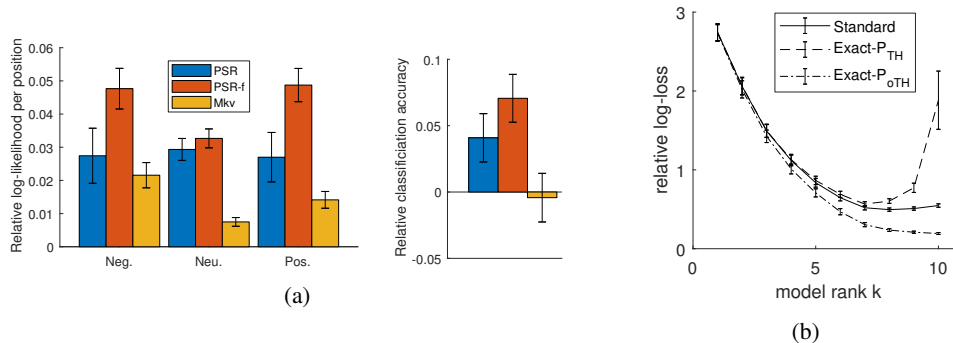

(a)                                                                                              (b)

Figure 2: **(a)** Results on the gene splice dataset. **Left:** relative log-likelihood (the higher the better) of learned models on in-class test sequences, where zero corresponds to a uniform model (all observations are i.i.d. and equally likely). **Right:** relative classification accuracy on test data (the higher the better), where zero corresponds to a classifier that always predicts the neutral label. **(b)** Unexpected results (Sec. 6.4). The figure shows the performance of standard spectral learning on RING HMMs, where certain empirical estimates are replaced by exact statistics. Sample size is 5000. See text for details.

$f(h) = [\hat{e} \quad \hat{e}^2 \quad \hat{l} \quad \hat{l}^2]^\top$. We generate $100, 200, \ldots, 500$ trajectories as training data, and evaluate the models on 1000 trajectories of length 3. $\mathcal{T}$ and $\mathcal{H}$ consist of all sequences of lengths up to 2 and 3, respectively.

Fig. 1b reports the log-loss of standard spectral learning, both using $f$ alone and using our Algorithm 3 as a function of sample size. Model rank is optimized separately between 1 and 20 for each model at each sample size. The figure shows that PSR-$f$ outperforms both PSRs and $f$ alone in the small sample region. As sample size grows, PSRs are able to improve by discovering good representations from the data, whereas using $f$ alone suffers from a fixed and limited representation. In this case, PSR-$f$ smoothly converges to match the PSR, enjoying the best of both worlds.

## 6.3 Gene splice dataset

Finally, we experiment on a gene splice dataset [13]. Each data point is a DNA sequence of length 60 and a class label that is either positive, negative, or neutral. The class prior is roughly 1:1:2. Following prior work [14], we train models of rank 4 for each class separately. Given models for each class, we use Bayes rule to compute the posterior over labels given the test sequence and predict the label that has the highest posterior. We compare different algorithms using the log-likelihood of test sequences from the same class, as well as using classification accuracy.

For PSR and PSR-$f$, $\mathcal{H}$ and $\mathcal{T}$ are set to all sequences up to length 4. We estimate the empirical statistics by a moving window approach to make full use of long sequences [15], which effectively turns every long sequence into 55 short sequences. We use 200 long sequences as training data, and 1000 sequences as test data. The PSR-$f$ learned from Algorithm 3 uses a simple $2^{\text{nd}}$ order Markov representation as $f$ (a one-hot vector with $m = 16$). The hyperparameter $d$ is tuned by 5-fold cross validation. As an additional baseline, we also learn a rank-4 model using $f$ as the state representation by first randomly projecting it down to 4 dimensions and then learning $\beta_o$ as in the synthetic experiments. We run this baseline 5 times and report the best performance (legend: "Mkv"). Fig. 2a shows the prediction accuracy on test sequences as well as the final classification accuracy. Again we observe that PSR-$f$ is able to outperform the standard PSR and the Markov baseline under both metrics, even when the domain knowledge provided by $f$ is fairly weak.

## 6.4 Unexpected results

We conduct further experiments to empirically explore what kind of $f$ is most beneficial. The full experiments and findings are deferred to Appendix G due to space limitations. Surprisingly, we find some highly counter-intuitive behavior that cannot be well explained by existing theory. Roughly speaking, giving away exact statistics to the algorithm can sometimes hurt performance drastically.

To show that this is not specific to our setting, we are able to produce similar behavior in the standard PSR learning setting without $f$. Fig. 2b shows the performance of the standard spectral algorithm and its 2 variants: (1) $\hat{P}_{\mathcal{T},\mathcal{H}}$ is replaced by $P_{\mathcal{T},\mathcal{H}}$, and (2) $\hat{P}_{o\mathcal{T},\mathcal{H}}$ is replaced by $P_{o\mathcal{T},\mathcal{H}}$. While we expect both variants to improve over the baseline, using exact $P_{\mathcal{T},\mathcal{H}}$ results in severely degenerate performance when model is full rank. More detailed discussions are found in Appendix G, and further investigations may lead to new theoretical and practical insights of spectral learning.

## 7 Conclusions

We proposed the PSR-$f$, a model that generalizes PSRs by taking advantage of a representation $f$ that encodes domain knowledge. Our Algorithm 3 spectrally learns PSR-$f$ models and discovers relevant components of $f$ using principal angles. The algorithm preserves the dimension of state, is invariant to transformation of $f$, and can achieve reduced model complexity when $f$ contains useful information. Future research directions include extending PSR-$f$ to allow more powerful regression tools, and unifying PSR-$f$ with prior work based on discrete-valued side information [5, 6, 11].

### Acknowledgments

This work was supported by NSF grant IIS 1319365. Any opinions, findings, conclusions, or recommendations expressed here are those of the authors and do not necessarily reflect the views of the sponsors.

## Footnotes

[1] See Appendix B for why we do not adopt the more popular $\{b_*, \{B_o\}, b_\infty\}$ parameterization.

[2]We describe the scenario where $f(h) = P_{\mathcal{T}_f|h}$ only to help readers transfer their knowledge from PSR to PSR-$f$. We expect that $f(h)$ will not take such a predictive format in practice. See the experiment setup in the aircraft domain for an example of a more natural choice of $f$.

[3]rowspace($P$) is the linear span of the row vectors of a matrix $P$.

[4]Strictly speaking, $\text{rank}(f)$ depends on $M$, which is implicit in notation. The slight dependence comes from the fact that $P_{f,\mathcal{H}} = f_{\mathcal{H}} \, \text{diag}(P_{\epsilon,\mathcal{H}})$ and $P_{\epsilon,\mathcal{H}}$ depends on $M$. The dependence, however, is minimal, since for any $\mathcal{H} \subset \mathcal{O}^*$ we have $\text{rank}(P_{f,\mathcal{H}}) = \text{rank}(f_{\mathcal{H}} \, \text{diag}(P_{\epsilon,\mathcal{H}})) = \text{rank}(f_{\mathcal{H}})$ as long as $P(h) \neq 0$, $\forall h \in \mathcal{O}^*$.

[5]Note that a mis-specified $f$ can still make learning $U_f$ more challenging when $m$ is large; however, the impact is much less significant than in Algorithm 2 or the alternative approaches discussed in Appendix E.1.

[6] It should be noted that learning a separate model for each partition enables nonlinear dependence on $f$, which cannot be directly expressed in our framework. However, Sec. 3 gives the regression view of PSR-$f$, which can be extended to nonlinear regression as done by Hefny et al. [8].

[7] The HMMs used to generate the irrelevant features have RAND topology, 20 states, 20 observations, 3 possible next states, and 20 possible observations per state.

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
