[Supplementary Material]

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

## Appendix

## A  Warm-up: Markov chain as a PSR

We show an example where $f$ lifted from a discrete-valued representation can be viewed as a predictive representation up to transformation. Consider a Markov chain over $n$ states $\mathcal{O} = \{1, \ldots, n\}$. Assume that the transition matrix $T \in \mathbb{R}^{n \times n}$ (current state on columns and next state on rows) is invertible. Let the initial distribution $\pi \in \mathbb{R}^n$ be a stationary distribution, that is, $T\pi = \pi$.

It is not hard to see that $\mathcal{T} = \mathcal{H} = \mathcal{O}$ are minimal core, so $P_{\mathcal{T}|h}$ is state. We can also have a transformed state $U^\top P_{\mathcal{T}|h}$ for any invertible matrix $U$ [16]. If we supply $U^\top = T^{-1}$ as the transformation matrix, the resulting state representation is $b(h) = U^\top P_{\mathcal{T}|h} = T^{-1} P_{\mathcal{T}|h}$. Note that $P_{\mathcal{T}|h}$ is simply the $o_{|h|}$-th column of $T$, where $o_{|h|}$ is the last observation of $h$. As a result, $T^{-1} P_{\mathcal{T}|h}$ is a unit vector with the $o_{|h|}$-th entry being 1, so $b(h)$ is the 1$^{\text{st}}$ order Markov representation.

## B  On Normalization Vectors $\{\beta_o\}_{o \in \mathcal{O}}$

Our PSR model uses observation-specific normalization vectors $\{\beta_o\}$, a parameterization that appeared in the early PSR literature [1]. Readers familiar with the recent spectral learning literature may be more comfortable with a single normalization vector $b_\infty$ [see e.g., 4]; it is in fact easy to translate between the two formulations by letting $\beta_o^\top = b_\infty^\top B_o$. We do not adopt $b_\infty$ because it creates problems for PSR-$f$: Suppose we use $b_\infty$ instead of $\beta_o$; then the prediction rule would be $P(o|h) \approx b_\infty^\top \begin{bmatrix} B_o b(h) \\ P(o|h) \cdot f(ho) \end{bmatrix}$. $P(o|h)$ now appears on the RHS, so we need to solve this equation to obtain $P(o|h)$, introducing an unnecessary complication. Moreover, when $b(\cdot) = f(\cdot)$ ($f$ gives the full state), the equation degrades to an identity (note that $b_\infty^\top f(ho) = 1$ if $f$ is a correct full state) and $P(o|h)$ cannot be determined. Using $\beta_o$ resolves this issue and makes the prediction rule much more straightforward.

## C  Additional Results and Proofs of Sec. 3

We first give some additional properties of the notions defined in Sec. 3. At the end we will use them to prove Theorems 1 and 2.

**Proposition 2** (Linear Relevance $\Leftrightarrow$ Predictive Representation). *For a system $M$ where all observation sequences have non-zero probabilities, $f$ is linearly relevant to $M$ if and only if $f(\cdot) = A^\top P_{\mathcal{T}_f|(\cdot)}$ for some tests $\mathcal{T}_f$ and transformation matrix $A^{|\mathcal{T}_f| \times m}$.*

*Proof.* "$\Leftarrow$" is trivial. To prove "$\Rightarrow$", we explicitly construct $\mathcal{T}_f$ and $A$: We choose $\mathcal{T}_f$ to be any core tests of $M$. Let $\mathcal{H}$ be any core histories. Due to linear relevance, rowspace$(P_{f,\mathcal{H}}) \subseteq$ rowspace$(P_{\mathcal{T}_f,\mathcal{H}})$, so we can find a matrix $A$ such that $P_{f,\mathcal{H}} = A^\top P_{\mathcal{T},\mathcal{H}}$. It then suffices to show that $f(h) = A^\top P_{\mathcal{T}_f|h}$, $\forall h \in \mathcal{O}^*$. Since all observation sequences have non-zero probabilities, this is equivalent to $P(h)f(h) = A^\top P_{\mathcal{T}_f,h}$. We show this by considering the following block matrix:

$$\begin{bmatrix} P_{\mathcal{T}_f,\mathcal{H}} & P_{\mathcal{T}_f,h} \\ P_{f,\mathcal{H}} & P(h)f(h) \end{bmatrix}.$$

$P_{\mathcal{T}_f,\mathcal{H}}$ is a full-rank submatrix of the block matrix and is invertible, so $P(h)f(h) = P_{f,\mathcal{H}} P_{\mathcal{T}_f,\mathcal{H}}^{-1} P_{\mathcal{T}_f,h} = A^\top P_{\mathcal{T}_f,h}$. This completes the proof. $\square$

**Proposition 3.** *For any $\mathcal{H}$ core w.r.t. $f$, (1) $\mathrm{rank}(P_{f,\mathcal{H}}) = \mathrm{rank}(f)$, and (2) $\mathcal{H}$ is also core in the usual sense.*

*Proof.* Let $\mathcal{T}$ be core w.r.t. $f$. For any $h \in \mathcal{O}^*$, $\begin{bmatrix} P_{\mathcal{T},h} \\ P_{f,h} \end{bmatrix}$ can be written as a linear combination of the column vectors of $\begin{bmatrix} P_{\mathcal{T},\mathcal{H}} \\ P_{f,\mathcal{H}} \end{bmatrix}$ since $\mathcal{H}$ is rank maximizing. This implies that $P_{f,h}$ is a linear

combination of the column vectors of $P_{f,\mathcal{H}}$, so adding any $h$ cannot increase the (column) rank of $P_{f,\mathcal{H}}$, hence $\mathrm{rank}(P_{f,\mathcal{H}}) = \mathrm{rank}(f)$. Similarly we can show that $\mathrm{rank}(P_{\mathcal{T},\mathcal{H}}) = \mathrm{rank}(M)$ if $\mathcal{T}$ is core in the usual sense, so $\mathcal{H}$ is also core in the usual sense. $\qquad\square$

**Proposition 4.** *Let $\mathcal{H}$ be any core histories of $M$. $\mathrm{rank}(P^{\star}_{f,\mathcal{H}}) = \mathrm{rank}(f; M)$.*

*Proof.* Let $\mathcal{H}'$ be the histories that achieve the supremum in Definition 3, that is $\mathrm{rank}(P^{\star}_{f,\mathcal{H}'}) = \mathrm{rank}(f; M)$. Let $\mathcal{T}$ be any core tests of $M$. Consider the block matrix

$$\begin{bmatrix} P_{\mathcal{T},\mathcal{H}} & P_{\mathcal{T},\mathcal{H}'} \\ P^{\star}_{f,\mathcal{H}} & P^{\star}_{f,\mathcal{H}'} \end{bmatrix}.$$

Since $P_{\mathcal{T},\mathcal{H}}$ is a full-rank submatrix of the above $2 \times 2$ block matrix,

$$\mathrm{rank}(f; M) = \mathrm{rank}(P^{\star}_{f,\mathcal{H}'}) = \mathrm{rank}(P^{\star}_{f,\mathcal{H}} P^{+}_{\mathcal{T},\mathcal{H}} P_{\mathcal{T},\mathcal{H}'})$$
$$\leq \min\{\mathrm{rank}(P^{\star}_{f,\mathcal{H}}), \mathrm{rank}(P^{+}_{\mathcal{T},\mathcal{H}}), \mathrm{rank}(P_{\mathcal{T},\mathcal{H}'})\} \leq \mathrm{rank}(P^{\star}_{f,\mathcal{H}}).$$

On the other hand, we have $\mathrm{rank}(P^{\star}_{f,\mathcal{H}}) \leq \mathrm{rank}(f; M)$ by definition, hence $\mathrm{rank}(P^{\star}_{f,\mathcal{H}}) = \mathrm{rank}(f; M)$. $\qquad\square$

**Definition 6** (Explicit construction of $U_f$ in Definition 4). *Let $\mathcal{T}$ be any core tests of $M$. Let $\mathcal{H}$ be core histories w.r.t. $f$. Let $V$ be a matrix whose rows form a basis of $rowspace(P_{f,\mathcal{H}}) \bigcap rowspace(P_{\mathcal{T},\mathcal{H}})$. Let $U_V$ be such that $V = U_V^{\top} P_{f,\mathcal{H}}$.*

**Proposition 5.** *$U_V$ in Definition 6 satisfies the conditions for $U_f$ in Definition 4.*

*Proof.* First, we show that appending any column vector $u$ to $U_V \in \mathbb{R}^{m \times n}$ will violate Definition 4: If $u^{\top} P_{f,\mathcal{H}} \notin rowspace(P_{\mathcal{T},\mathcal{H}})$, $[U_V\ u]^{\top} f(\cdot)$ is not linearly relevant. On the other hand, if $u^{\top} P_{f,\mathcal{H}} \in rowspace(P_{\mathcal{T},\mathcal{H}})$, then it also lies in $rowspace(P_{\mathcal{T},\mathcal{H}}) \bigcap rowspace(P_{f,\mathcal{H}})$. However, $U_V^{\top} P_{f,\mathcal{H}}$ is already a basis for that space, so $u^{\top} P_{f,\mathcal{H}}$ is not linearly independent of $U_V^{\top} P_{f,\mathcal{H}}$. As a result, $\mathrm{rank}(U_V^{\top} f(\cdot)) = n \leq n + 1$, which violates requirement (2) in Definition 4.

It then suffices to prove linear relevance. That is, for any $H' \subset \mathcal{O}^*$, $rowspace(U^{\top} P_{f,\mathcal{H}}) \subseteq rowspace(P_{\mathcal{T},\mathcal{H}})$ for any core $\mathcal{T}$ of $M$. To prove this, pick $\mathcal{T}' \subset \mathcal{T}$ such that rows of $\begin{bmatrix} P_{\mathcal{T}',\mathcal{H}} \\ V \end{bmatrix}$ form a basis of $rowspace(P_{\mathcal{T},\mathcal{H}})$. Similarly pick rows of $P_{f,\mathcal{H}}$ indexed by $I$, denoted as $P_{f^I,\mathcal{H}}$, such that rows of $\begin{bmatrix} V \\ P_{f^I,\mathcal{H}} \end{bmatrix}$ form a basis of $rowspace(P_{f,\mathcal{H}})$. Note that (1) the rows of $P_{\mathcal{T}',\mathcal{H}}$, $P_{f^I,\mathcal{H}}$ and $V$ are linearly independent, (2) $P_{\mathcal{T}',\mathcal{H}}$ are linearly independent of all the rows of $P_{f,\mathcal{H}}$, and (3) $P_{f^I,\mathcal{H}}$ are linearly independent of all the rows of $P_{\mathcal{T},\mathcal{H}}$, otherwise $V$ could have more rows.

Now let $\mathcal{H}^+ = \mathcal{H} \bigcup \mathcal{H}'$, and $V' := U_V^{\top} P_{f,\mathcal{H}^+}$. Since $V$ is submatrix of $V'$, all the linear independence statements transfer to $P_{\mathcal{T}',\mathcal{H}^+}$, $P_{f^I,\mathcal{H}^+}$ and $V'$. Assume towards contradiction that exactly one row of $V'$, denoted as $v'$, does not lie in $rowspace(P_{\mathcal{T},\mathcal{H}^+})$. (The proof can be extended to arbitrary number of rows.) We can find $t \in \mathcal{T} \setminus \mathcal{T}'$ such that $P_{t,\mathcal{H}^+}$, $P_{\mathcal{T}',\mathcal{H}^+}$, and the remaining $(n-1)$ rows of $V'$ form a basis of $rowspace(P_{\mathcal{T},\mathcal{H}^+})$. Since $v'$ does not lie in this space, it is linearly independent of all three of them. Furthermore, $P_{f^I,\mathcal{H}^+}$ are independent of all of $V'$ as well as all rows of $P_{\mathcal{T},\mathcal{H}^+}$. All together we have that the rows of $P_{t,\mathcal{H}^+}$, $P_{\mathcal{T}',\mathcal{H}^+}$, $P_{f^I,\mathcal{H}^+}$ and $V'$ are linearly independent, implying that $\mathrm{rank}\left(\begin{bmatrix} P_{\mathcal{T},\mathcal{H}^+} \\ P_{f,\mathcal{H}^+} \end{bmatrix}\right) = \mathrm{rank}\left(\begin{bmatrix} P_{\mathcal{T},\mathcal{H}} \\ P_{f,\mathcal{H}} \end{bmatrix}\right) + 1$. This contradicts the fact that $\mathcal{H}$ is core w.r.t. $f$ and has maximized the rank of $\begin{bmatrix} P_{\mathcal{T},\mathcal{H}} \\ P_{f,\mathcal{H}} \end{bmatrix}$. $\qquad\square$

**Proposition 6.** *Let $(\mathcal{T}, \mathcal{H})$ be those in Definition 6,*

$$\mathrm{rank}(f; M) = \dim(rowspace(P_{f,\mathcal{H}}) \bigcap rowspace(P_{\mathcal{T},\mathcal{H}})).$$

*Proof.* From Prop. 3, $\mathcal{H}$ is core to $M$. From Prop. 4 and Prop. 5,

$$\mathrm{rank}(f; M) = \mathrm{rank}(P^{\star}_{f,\mathcal{H}}) = \mathrm{rank}(U_V^{\top} P_{f,\mathcal{H}})$$
$$= \mathrm{rank}(V) = \dim(rowspace(P_{f,\mathcal{H}}) \bigcap rowspace(P_{\mathcal{T},\mathcal{H}})). \qquad\square$$

***Proof of Theorem 1***. It suffices to show two things: (1) Eq.(5) has an exact solution (which is found by matrix inverse), and (2) the solution to Eq.(5) predicts $P(o|h)$ and makes state update correctly for any $h \in \mathcal{O}^*$, even if $h \notin \mathcal{H}$. Given any $h \in \mathcal{O}^*, o \in \mathcal{O}$, consider the following block matrix:

$$\begin{bmatrix} P_{\mathcal{T},\mathcal{H}} & P_{\mathcal{T},h} \\ P_{f,\mathcal{H}} & P_{f,h} \\ P_{o\mathcal{T},\mathcal{H}} & P_{o\mathcal{T},h} \\ P_{o,\mathcal{H}} & P_{o,h} \end{bmatrix}. \tag{8}$$

Since $\begin{bmatrix} P_{\mathcal{T},\mathcal{H}} \\ P_{f,\mathcal{H}} \end{bmatrix}$ is its full-rank submatrix (Definition 1), there exists matrices $B_o$ and $\beta_o$ that exactly transform $\begin{bmatrix} P_{\mathcal{T},\mathcal{H}} \\ P_{f,\mathcal{H}} \end{bmatrix}$ to $P_{o\mathcal{T},\mathcal{H}}$ and to $P_{o,\mathcal{H}}$, respectively, which proves (1). Furthermore, the same matrices will transform $\begin{bmatrix} P_{\mathcal{T},h} \\ P_{f,h} \end{bmatrix}$ to $P_{o\mathcal{T},h}$ and $P_{o,h}$, which proves (2). $\square$

***Proof of Theorem 2***. Since $(\mathcal{T}, \mathcal{H})$ in Definition 6 is rank maximizing, we can compute the row rank of $\begin{bmatrix} P_{\mathcal{T},\mathcal{H}} \\ P_{f,\mathcal{H}} \end{bmatrix}$ to obtain the minimal size of $\mathcal{H}$: $\mathrm{rank}(P_{\mathcal{T},\mathcal{H}}) = \mathrm{rank}(M), \mathrm{rank}(P_{f,\mathcal{H}}) = \mathrm{rank}(f)$ (Prop. 3), and adding them together double counts the dimension of the joint row space, $\mathrm{rank}(f; M)$ (Prop. 6), hence the rank is $\mathrm{rank}(M) + \mathrm{rank}(f) - \mathrm{rank}(f; M)$ and so is the minimal size of core $\mathcal{H}$ w.r.t. $f$. The minimal size of core tests w.r.t. $f$ can be computed via the constructive process given in the proof of Prop. 5: $\mathcal{T}'$ there is a minimal core test set and $|\mathcal{T}'| = \mathrm{rank}(M) - \mathrm{rank}(V) = \mathrm{rank}(M) - \mathrm{rank}(f; M)$. $\square$

We also give an alternative definition of $\mathrm{rank}(f; M)$ below.

**Proposition 7.** $\mathrm{rank}(f; M) = \mathrm{rank}(M) - \sup_{\mathcal{T} \subset \mathcal{O}^*, \mathcal{H} \subset \mathcal{O}^*} \left( \mathrm{rank}\left( \begin{bmatrix} P_{\mathcal{T},\mathcal{H}} \\ P_{f,\mathcal{H}} \end{bmatrix} \right) - \mathrm{rank}(P_{f,\mathcal{H}}) \right).$

*Proof.* We prove that $\mathrm{rank}\left( \begin{bmatrix} P_{\mathcal{T},\mathcal{H}} \\ P_{f,\mathcal{H}} \end{bmatrix} \right) - \mathrm{rank}(P_{f,\mathcal{H}})$ is a non-decreasing function of $\mathcal{H}$. Once we prove this, $(\mathcal{T}, \mathcal{H})$ achieves the supremum in the theorem statement when $\mathcal{T}$ is core to $M$ and $\mathcal{H}$ is rank maximizing as in Definition 6. From the proof of Theorem 2 it is clear that $\mathrm{rank}(\begin{bmatrix} P_{\mathcal{T},\mathcal{H}} \\ P_{f,\mathcal{H}} \end{bmatrix}) = \mathrm{rank}(M) + \mathrm{rank}(f) - \mathrm{rank}(f; M)$ and $\mathrm{rank}(P_{f,\mathcal{H}}) = \mathrm{rank}(f)$.

To prove the non-decreasing property, consider any $\mathcal{H} \subset \mathcal{O}^*$ and we add a history $h$ to it. We show that if this increases the rank of $P_{f,\mathcal{H}}$ it must also increase the rank of the concatenated matrix. This is true because, if the new column $P_{f,h}$ is linearly independent of $P_{f,\mathcal{H}}$, then its extended vector $\begin{bmatrix} P_{\mathcal{T},h} \\ P_{f,h} \end{bmatrix}$ is also linearly independent of $\begin{bmatrix} P_{\mathcal{T},\mathcal{H}} \\ P_{f,\mathcal{H}} \end{bmatrix}$. $\square$

### C.1 Example that illustrates why $|\mathcal{H}| > \mathrm{rank}(M)$ can be necessary

Suppose $\mathcal{H}$ is core in the usual PSR sense ($|\mathcal{H}| = \mathrm{rank}(M)$). Consider two different functions $f_1$ and $f_2$, where $f_1$ is linearly relevant. $f_2$, however, agrees with $f_1$ on every $h \in \mathcal{H}$, but makes irrelevant predictions on $h \notin H$. If we solve Eq.(5) with such a $\mathcal{H}$, the resulting model parameters are the same for $f_1$ and $f_2$, but the model only makes correct predictions with $f_1$ and apparently errs with $f_2$. The reason is that $f_2$ has different behavior on $h \in \mathcal{H}$ and $h \notin \mathcal{H}$, and the learning algorithm needs to look at a larger set of histories to evidence such inconsistency and avoid erroneously relying on a bad function.

## D   On Feature PSRs

We give more details on why history features in Feature PSRs [4] are very different from our use of $f$, despite that both are functions that map histories to $\mathbb{R}^m$. Vanilla PSRs may be viewed as

Feature PSRs with indicator features of $\mathcal{H}$ and $\mathcal{T}$. In this case, the history features can be written as $f(h) = [\mathbb{I}[h = h']]_{h' \in \mathcal{H}}$. While such $f$ is a good choice as history features as long as $\mathcal{H}$ is core, it is a terrible choice for our purpose, as $f(h) = \mathbf{0}$ for all $h \notin \mathcal{H}$, and in general a good $f$ for our purpose should provide informative values for all observation sequences (c.f. Appendix C.1). Our empirical results in Sec. 6.3 also shows that PSR-$f$ with $f$ being the $2^{\text{nd}}$ order Markov representation can outperform PSRs, although all length 2 observation sequences are "already" in $\mathcal{H}$. This shows that the two approaches are complementary and require different kinds of functions of histories to work well.

# E  Additional Results and Proofs of Sec. 4

***Proof of Theorem 3****.* The proof is almost identical to that of Theorem 1, and we only need to show that if we replace $\begin{bmatrix} P_{\mathcal{T},\mathcal{H}} \\ P_{f,\mathcal{H}} \end{bmatrix}$ with $U^{\top} \begin{bmatrix} P_{\mathcal{T},\mathcal{H}} \\ P_{f,\mathcal{H}} \end{bmatrix}$ in Eq.(8), where $U = \begin{bmatrix} U_{\mathcal{T}} \\ U_f \end{bmatrix}$ comes from the algorithm, $U^{\top} \begin{bmatrix} P_{\mathcal{T},\mathcal{H}} \\ P_{f,\mathcal{H}} \end{bmatrix}$ is still a full-rank submatrix of the $2 \times 2$ block matrix. This is true because $\text{rank}(\begin{bmatrix} P_{\mathcal{T},\mathcal{H}} \\ P_{f,\mathcal{H}} \end{bmatrix}) = \text{rank}(M) + \text{rank}(f) - \text{rank}(f; M)$ (see proof of Theorem 1), and $U$ is the leading left singular vectors of $\begin{bmatrix} \hat{P}_{\mathcal{T},\mathcal{H}} \\ \hat{P}_{f,\mathcal{H}} \end{bmatrix}$. When $\begin{bmatrix} \hat{P}_{\mathcal{T},\mathcal{H}} \\ \hat{P}_{f,\mathcal{H}} \end{bmatrix}$ converges to $\begin{bmatrix} P_{\mathcal{T},\mathcal{H}} \\ P_{f,\mathcal{H}} \end{bmatrix}$, $U$ preserves the rank of $\begin{bmatrix} P_{\mathcal{T},\mathcal{H}} \\ P_{f,\mathcal{H}} \end{bmatrix}$, so the argument in the proof of Theorem 1 still applies. $\qquad\square$

***Proof of Theorem 4****.* We use the same argument as before and show that $U^{\top} \begin{bmatrix} P_{\mathcal{T},\mathcal{H}} \\ P_{f,\mathcal{H}} \end{bmatrix}$ is a full-rank submatrix of Eq.(8). First notice that when $\hat{P}_{\mathcal{T},\mathcal{H}}$ and $\hat{P}_{f,\mathcal{H}}$ converges to the true statistics and $(\mathcal{T}, \mathcal{H})$ is core w.r.t. $f$, the $\text{rank}(f; M)$ smallest principal angles are all 0's, and correspondingly, the $\text{rank}(f; M)$ largest singular values are all 1's. Therefore, taking the $d = \text{rank}(f; M)$ largest singular values on Line 4 identifies all the linearly relevant components of $f$. When Algorithm 2 is called as a subroutine, the new function $f' = \lambda(U')^{\top} f$ is linearly relevant. When $\lambda > 0$ and $(\mathcal{T}, \mathcal{H})$ is core w.r.t. $f$, the row space of $\begin{bmatrix} P_{\mathcal{T},\mathcal{H}} \\ P_{f',\mathcal{H}} \end{bmatrix}$ is equal to that of $P_{\mathcal{T}',\mathcal{H}}$ for core tests $\mathcal{T}'$ (in the usual sense). Therefore, truncating the SVD at $k = \text{rank}(M)$ still results in a full-rank submatrix of Eq.(8). The remainder of the proof follows the same argument as that of Theorem 1. $\qquad\square$

## E.1  On the choice of $\lambda$ in Algorithm 3

Before proving Prop. 1, we use a simple example to illustrate the effect of $\lambda$. Consider the following matrix:

$$\begin{bmatrix} a \\ \lambda a \end{bmatrix} \tag{9}$$

where $a \in \mathbb{R}$ is some non-zero scalar and $\lambda \geq 0$. This simple example is an analogy of $\begin{bmatrix} P_{\mathcal{T},\mathcal{H}} \\ P_{\lambda f',\mathcal{H}} \end{bmatrix}$ as in Algorithm 3. The idea is that $a$ resembles $P_{\mathcal{T},\mathcal{H}}$, and $\lambda a$ resembles $P_{f',\mathcal{H}}$, where $f' = (\lambda U')^{\top} f$ is the linearly relevant components of $f$ rescaled by $\lambda$. The first left singular vector of Eq.(9) is

$$\begin{bmatrix} \frac{1}{\sqrt{\lambda^2 + 1}} \\ \frac{\lambda}{\sqrt{\lambda^2 + 1}} \end{bmatrix}.$$

So the learned "state representation" is a combination of "$P_{\mathcal{T}|h}$" (corresponding to $a$) and "$f$" (corresponding to $\lambda a$). In general, the greater $\lambda$, the more heavily the learned representation relies on $f$. In this particular case, $f$ contains all the information in $P_{\mathcal{T},\mathcal{H}}$ and is accurate, while $P_{\mathcal{T},\mathcal{H}}$ is generally prone to statistical errors. Clearly we would like to use $f$ alone as the state representation, which is achieved by letting $\lambda \to \infty$.

Figure 3: Full results on synthetic HMMs. Fig. 1a corresponds to the second column here.

The above argument shows that $\lambda \to \infty$ allows us to fully trust the information in $(U')^\top f$ and avoid using the equally informative yet noisy statistics from $P_{\mathcal{T},\mathcal{H}}$. In general, however, if we just rescale $f$ by $\lambda$ without extracting out the relevant components first, then we are also putting a heavy weight on the irrelevant components, causing the blow-up of state dimensionality described in Sec. 4.1.

***Proof of Prop. 1.*** Consider the SVD in Algorithm 2 when it is called as a subroutine in Algorithm 3. Since $\lambda \to \infty$ causes the spectrum to blow up to $\infty$, we instead study the spectrum of $\frac{1}{\lambda} \begin{bmatrix} \hat{P}_{\mathcal{T},\mathcal{H}} \\ \hat{P}_{f',\mathcal{H}} \end{bmatrix}$, which has the same decomposition up to a rescaling of singular values. As $\lambda \to \infty$,

$$\frac{1}{\lambda} \begin{bmatrix} \hat{P}_{\mathcal{T},\mathcal{H}} \\ \hat{P}_{f',\mathcal{H}} \end{bmatrix} \to \begin{bmatrix} \mathbf{0}_{|\mathcal{T}| \times |\mathcal{H}|} \\ \hat{P}_{U'f,\mathcal{H}} \end{bmatrix}.$$

From Algorithm 3 we know that $\hat{P}_{U'f,\mathcal{H}}$ has orthonormal rows, so the spectrum of the RHS of the above equation is $\{1, \ldots, 1, 0, \ldots, 0\}$, where there are $d$ leading 1's (recall that $d$ is the dimensionality of $U'f$). The associated left singular vectors $u_1, \ldots, u_d$ take the form of

$$u_i = \begin{bmatrix} \mathbf{0}_{|\mathcal{T}|} \\ u'_i \end{bmatrix}, \; i = 1, \ldots, d,$$

where $u'_i \in \mathbb{R}^d$ is the non-zero component of $u_i$. Algorithm 2 then assigns $U_{\mathcal{T}} := U_{1:|\mathcal{T}|,1:k}$. Since $d \le k$, the first $d$ columns of $U_{\mathcal{T}}$ consist of the upper halves (with row indices 1 to $|\mathcal{T}|$) of $u_1, \ldots, u_d$, which are all zeros. This directly translates into a $B_o$ matrix whose first $d$ rows are all zeros (see Algorithm 1), hence the number of non-zero entries in $B_o$ is at most $k(k-d)$. $\qquad\square$

## F    Additional Experiment Details

**Model Predictions and Rectifications**    When predicting $P(x)$ for sequence $x = o_1 \ldots o_{|x|} \in \mathcal{O}^*$, we always break it down to the product of conditional probabilities

$$P(x) = P(o_1)P(o_2 \mid o_1)P(o_3 \mid o_1 o_2) \ldots P(o_{|x|} \mid o_1 \ldots o_{|x|-1}).$$

So a model only needs to have the ability to predict $P(o|h)$ for any $o \in \mathcal{O}$ and $h \in \mathcal{O}^*$. It is known, however, that PSR models can yield negative and/or unnormalized probability predictions [14, 17], so

we rectify and normalize the conditional prediction as follows: given model predictions $[\hat{P}(o|h)]_{o\in\mathcal{O}}$, we first convert it to $[\max\{10^{-3}, \hat{P}(o|h)\}]_{o\in\mathcal{O}}$ and then normalize it. This guarantees that the model prediction induces a valid and consistent distribution over observation sequences.

**Details of Aircraft Identification domain** There are 22 observations in total. 2 of them are special observations, and the remaining 20 correspond to foe/friend signal and a discrete distance between 0 and 9. When we compute $\hat{e}$ and $\hat{l}$, we ignore any special observations in the history and average the signals and distances among the remaining time steps.

When optimizing model rank, we first performed a grid search over $\{2, 4, \ldots, 20\}$, which reveals that all methods start over-fitting when model rank $\geq 6$ under the sample sizes of interest. In the final result we searched over ranks $\{1, 2, 3, 4, 5\}$ and reported the best performance for each method under each sample size.

**Implementation of "f-only"** For synthetic HMMs and the aircraft identification domain, we implement the "f-only" baseline by calling Algorithm 2 with model rank $k = m$ (recall that $m$ is the dimension of $f$) after rescaling $f$ to $10^4 f$. This ensures that SVD picks up and only picks up $\hat{P}_{f,\mathcal{H}}$ as its representation. For the gene splice experiment, since we use a moving window approach for PSRs, we also change the implementation of "f-only" accordingly: for each point in the sequence, we extract $h$ and $o$ where $h$ is the history at that point and $o$ is the immediate next observation, and form an input-output pair $(f(h), [\mathbb{I}\{o' = o\}]_{o'\in\mathcal{O}})$, and run linear regression on such pairs to learn $\{\beta_o\}$.

# G    Negative Results and Implications

Our theory characterizes the usefulness of $f$ by linear relevance. When two functions are both relevant, however, they may still have different degrees of usefulness in practice, and we conduct further experiments to empirically explore more fine-grained notions of usefulness. The results are surprising and lead to questions on our theoretical understanding of spectral learning.

Figure 4: Additional results on RING HMMs. $f = u_i^\top P_{\mathcal{T}|h}$ where $u_i$ is the $i$-th left singular vector of $P_{\mathcal{T},\mathcal{H}}$. X-axis shows $i$. Horizontal line shows the test loss of standard PSRs. Vertical dotted lines mark the model rank. All results are averaged over $50$ trials.

The experiment setting is mostly the same as Sec. 6.1; here we only report results on RING HMMs, but the phenomenon is agnostic to topology. The HMMs have 10 states, 20 observations, and 4 possible observations per state. For each HMM, we design multiple 1-dimensional functions $f_i = u_i^\top P_{\mathcal{T}|h}$, where $\mathcal{T}$ is the tests used in Sec. 6.1, and $u_i$ is the singular vector corresponding to the $i$-th largest singular value of $P_{\mathcal{T},\mathcal{H}}$, denoted as $\sigma_i$. We run Algorithm 3 with $d = 1$, forcing the algorithm to fully trust $f_i$, essentially giving away the $i$-th dimension of the state representation that would have been identified by spectral learning with exact statistics.

Fig. 4 shows the test loss of using different $f_i$ as a function of $i$ under different model ranks (columns) and different sample sizes (rows). In most plots the curve first decreases as $i$ increases from 1 to 2. This makes sense because the identification of the largest singular value is relatively stable, so $f$ may help more by providing the dimension that corresponds to a smaller singular value. For model rank $k \leq 8$, the curves start to increase after $i > k$ and become worse than the baseline of standard PSRs. This again makes some sense, because the low-rank model has limited capacity, and we are forcing it to include a state dimension that is less important than what it could include without $f$.[8]

The results, however, are counter-intuitive for $k = 10$, where model rank is equal to the number of hidden states (full-rank learning). In this case we still see a drastically degenerated performance for $i = 10$. Most existing theory for spectral learning is based on matrix perturbation analyses, which suggests that the smallest singular values are most difficult to identify, and thus they determine the finite sample behavior of spectral algorithms [3, 18, 19]. When $f_k$ is given and incorporated, the smallest singular value that the algorithm needs to identify from data effectively increases from $\sigma_k$ to $\sigma_{k-1}$. Theory implies that $f_k$ should make learning easier, whereas the empirical results show the polar opposite.

To verify that this issue is not specific to our algorithm, we conduct an additional experiment where $\hat{P}_{\mathcal{T},\mathcal{H}}$ is replaced with the exact statistics in the standard PSR algorithm. This is similar to the previous experiment in the sense that we give away the state representation. Standard analyses for spectral learning bound the errors $\|\hat{P}_{\mathcal{T},\mathcal{H}} - P_{\mathcal{T},\mathcal{H}}\|$ and $\|\hat{P}_{o\mathcal{T},\mathcal{H}} - P_{o\mathcal{T},\mathcal{H}}\|$ separately (see e.g., Lemma 8 in [3]), which implies that improving the accuracy of one of them while keeping the others intact should improve performance.

The theoretical intuitions are proved wrong once again: Fig. 2b in the main text shows the log-loss of the standard algorithm and the algorithm with exact $P_{\mathcal{T},\mathcal{H}}$ statistics. The latter has significantly worse performance when model rank is 10. We also note that this phenomenon is agnostic to HMM topology (e.g., RAND), the size of $\mathcal{H}$ and $\mathcal{T}$ (e.g., using larger sets that include length 3 sequences does not change the phenomenon qualitatively), and whether empirical or exact statistics of $P_{\mathcal{T},\epsilon}$ and $P_{o,\mathcal{T}}$ are used. While larger sample size will asymptotically eliminate any error, we find that the gap between standard spectral learning and its variant with exact $P_{\mathcal{T},\mathcal{H}}$ shrinks at a very slow rate.

One plausible explanation is that $\hat{P}_{\mathcal{T},\mathcal{H}}$ and $\hat{P}_{o\mathcal{T},\mathcal{H}}$ are consistent with each other in a certain sense, e.g., when $\mathcal{T} = \mathcal{H} = \mathcal{O}^*$, the latter is a submatrix of the former, and replacing one of them with exact statistics breaks this desirable property [15, 20]. The 3$^{\text{rd}}$ curve on the figure invalidates this explanation and makes things even more confusing: we replace $\hat{P}_{o\mathcal{T},\mathcal{H}}$ with the exact statistics while keeping $\hat{P}_{\mathcal{T},\mathcal{H}}$ intact, and see improved performance across all model ranks. Overall, this set of results reflect the looseness of our current analyses and our understanding of the theoretical aspects of spectral learning. We leave further investigation of this issue to future work.