[Reviews · NeurIPS 2018]

Reviewer 1



SUMMARY: This paper proposes a method to incorporate prior knowledge into the spectral learning algorithm for predictive state representations (PSR). The prior knowledge consists of an imperfect/incomplete state representation which is 'refined' and 'completed' by the learning algorithm. This contribution addresses one of the main caveats of spectral methods: while these methods are fast and consistent, they tend to perform worse than local methods (e.g. EM) in the low data regime. By leveraging domain specific knowledge, the proposed algorithm overcomes this issue. The proposed extension, PSR-f, is relatively straightforward: the belief vector at each time step is the concatenation of the user-specified state representation f with a learned state representation b; the parameters of b are learned in the same fashion as for the classical method by solving linear regression problems constrained to the row space of the concatenation of some Hankel/system matrices (e.g. now mapping [P(T | h) ; P(h)f(h)] to P(oT | h) for each B_o). The authors then give sufficient conditions for the algorithm to be consistent and formally analyze to which extent providing prior information to the algorithm can be beneficial or detrimental; indeed, the number of states of a minimal PSR-f can be greater than the one of the corresponding minimal PSR when f is 'irrelevant'! To circumvent this issue, the authors proposed a method to extract the relevant information from f before performing the spectral learning algorithm by analyzing the principal angles between the classical Hankel matrix and some feature matrix induced by the user-specified state representation f. The proposed method is evaluated on both synthetic and real data experiments where PSR-f is compared with the basic PSR spectral learning algorithm (i.e. not using the prior knowledge of f) and with only using the user-specified state representation f. OPINION: I liked the paper, it is well written and relatively easy to follow for someone familiar with the spectral learning literature. As mentioned above, I think that the problem addressed in the paper is relevant and the solution proposed by the author is natural and sound (I did not check the maths thoroughly but the results stated in the Theorems seems accurate and expected), and that this contribution is a nice addition to the spectral learning literature. The paper could maybe benefit from a more high-level / abstract exposition of the technical material, e.g. reasoning also in terms of subspaces rather than only with the Hankel sub-blocks in Defs 2 to 5 (see my comment below), but this may be a matter of personal taste. REMARKS/QUESTIONS: - I find it strange to introduce the \beta_o as parameters, is it standard notation? I am more familiar with having a normalizer beta_\infty as a parameter of the model instead and defining the update equation with \beta_\infty B_o b(h) at the denominator (see e.g. [4]). How are the two related? - Is the requirement that the core tests and histories are minimal necessary for the consistency results? - In definitions 2 and 3, I believe that H (and even T) can be replaced directly with O^* instead of having a universal quantifier and a sup respectively. Then, in my opinion, defining rank(f;M) as the dimension of the intersection of the row spaces of P_{O^*,O^*} and P_{f,O^*} is simpler and more intuitive (this is actually what is indirectly done in Prop. 7 in the supplementary). Indeed, this intersection is the smallest subspace of P_{f,O^*} that is contained in P_{O^*,O^*}, i.e. that is linearly relevant to M. Then the matrix U_f in definition 4 can be understood as some projection onto this intersection (i.e. the rows of U_f^T U_f P_{f,H} belongs both to the row spaces of P_{O^*,H} and P_{f,H}). Finally, Theorem 2 simply follows from the formulas for the dimension of the complement of P_{O^*,H} \cap P_{f,H} in P_{O^*,H} (for |T|), and for the dimension of the sum P_{O^*,H} + P_{f,H} (for |H|). Now, similarly to classical PSR, the minimal sizes for T and H are equal to rank of the concatenation of P_{O^*,H} and P_{f,H}, which can be as low as rank(M) but as high as rank(M) + n (when the intersection of P_{O^*,H} and P_{f,H} is {0}). - I find the phrasing "solving by matrix inverse" a bit odd. MINOR COMMENTS/TYPOS: - line 52: since there are no restriction on Z line 48, setting Z = R^n and assuming that P(o|h) is linear in the belief is actually a restriction rather than a generalization. - line 76: R -> R^m - line 113-114: use (resp. H) ... (reps. T). - line 136: spurious ) after H - line 158: paramter -> parameter

Reviewer 2



This paper proposes a method to include domain knowledge within the Predictive State Representation framework for learning dynamics model. Essentially, the PRS is augmented with a user defined history->state function. The paper shows how to shows how this user defined partial state reduces the model size. The paper seems technically sound, is well written and makes a modest but clear contribution to the field. For comments (in no particular order), see below. ================= QUALITY ======================== - The explanation of the aircraft experiment does not contain what space the distances comes from. This would be helpful to understand the size of the paper. - The explanation of the increased state dimensionality for the Algorithm in section 4.1 is strange, as the result in Theorem 3 does not depend on ||f(h)||. While the extreme case is a useful thought experiment, and probably helpful, the text as written is misleading. - The interpretation of the results should more clearly state which results are shown in Figure 1, and which must be looked up in the appendix. Particularly for the low-data-regime statements. - A discussion of the limitations and suggestions for further research is missing. There is some discussion in Appendix F, but this is a) incomplete in terms of presentation, i.e., omission of details and not as well written b) not referred to in the text. Given the importance of what was found, both these issues should be adressed, preferably with a small discussion added in the main body. - The different choices for f(h) in the experiments do not inspire confidence that the approach can use easy-to construct f(h) in many settings. While it is shown that the approach learns to neglect useless parts of f(h), further discussion/references about useful f(h) would be beneficial. ================= CLARITY ======================== - Compliments for the readability of this rather technical paper. The order of presentation, and the discussion/interpretation of the theorems are very helpful. - The use of a numerical footnote-symbol in definition 2 and 3 is better avoided. - Appendix B1 can be incorporated in the text. - The sentence "smoothly converges to match PSR" seems backwards (PSR approaches the PSR-f performance as sample size increases) . ================= ORIGINALITY/SIGNIFICANCE======== - The contribution of this paper is clear: extending the PSR-approach to include prior information in the form of a partial state. - The authors do not make clear to whom the original idea for their PSR-formulation (Section 3) belongs. Some further references to proofs mentioned in that section would also be helpful. == ERRATA== page 8: componens

Reviewer 3



The paper formulates a way to incorporate side information about the hidden state space to learn the predictive state representation more efficiently. As a result, an augmented PSR framework is defined along with the adapted spectral learning approach, backing these up with multiple theoretical statements (about dimensions, consistency, etc.). The side information can be imperfect, and it is taken care of by using only linear projections of this information that are mostly aligned with the empirical evidence. Several experiments are performed highlighting the fact that simple but imperfect side information can improve the performance of the learned model, compared to the standard spectral learning for PSRs. Clarity: The paper is well written. There are several aspects that are worth clarifying though: 1. Example that follows Eq. 4: It is useful to highlight that the assumption of knowing some independent tests means also knowing the values of those tests for any history (not merely knowing what some core tests are). 2. There is a lack of clarity for what makes “f” useful enough or not for learning. Lines 137-143 mention that “f” that is too irrelevant may be detrimental to learning due to large core f-histories. So, should one pick “f” whose core histories are within the dimensions of the estimated P_TH, for example? 3. Although some discussion is provided (mainly in supp material) about lambda in algorithm 3, it still feels like this is an additional parameter that should be optimized, possibly through cross validation. Is it the case? If not, a better justification is missing. Originality: This is new material as far as I know that is sufficiently different from related work. Quality: All the proofs appear in the supplementary material, so I didn’t check all of them carefully, but the statements make sense. The experiments appear to be done correctly as well. My only small concern is about the “Mkv” baseline in the gene splice dataset: a more convincing baseline is taking 4 most important projections based on SVD (of the 16dim matrix) instead of 4 random projections. This would correspond to a rank-4 PSR compression of the 16 state markov chain. Ideally, PSR-f should perform better than this as well. Also, good job on including the discussion on the “negative results” in the supplementary material, this is a valuable piece of information that requires further investigation. Importance: Finding ways to incorporate domain knowledge easily in the PSR framework is a meaningful direction to study and could result in learning better representations. A better understanding of the limitations and advantages of this approach, as well as what kind of functions “f” are useful is required to have stronger impact.